# Life Histories and Functional Responses of Two Predatory Mites Feeding on the Stored-Grain Pest *Liposcelis bostrychophila* Badonnel (Psocoptera: Liposcelididae)

**DOI:** 10.3390/insects14050478

**Published:** 2023-05-19

**Authors:** Weiwei Sun, Liyuan Xia, Yi Wu

**Affiliations:** Academy of National Food and Strategic Reserves Administration, National Engineering Research Center for Grain Storage and Transportation, No. 11 Baiwanzhuang Street, Beijing 100037, China; sunweiw@email.swu.edu.cn (W.S.); xly@ags.ac.cn (L.X.)

**Keywords:** *Cheyletus malaccensis*, *Cheyletus eruditus*, biological control, predation efficiency

## Abstract

**Simple Summary:**

To improve the control of stored-grain pests and protect the environment, the use of natural enemies is attracting increasing attention. *Cheyletus malaccensis* and *Cheyletus eruditus* are effective natural enemies of common stored-grain pests, such as *Liposcelis bostrychophila* Badonnel. To evaluate the predation potential of these two types of predatory mites, this study was conducted by comparing the life histories of two predatory mites and the functional responses on *Liposcelis bostrychophila*. The results indicate that compared to *C. eruditus*, *C. malaccensis* had a shorter development time and longer adult survival time at 28 °C and 75% RH and showed higher predation ability against *Liposcelis bostrychophila* eggs. It has been proven from both its life history of artificial breeding and the predation ability that *C. malaccensis* has much greater biocontrol potential than *C. eruditus* against stored-grain pests.

**Abstract:**

*Cheyletus malaccensis* Oudemans and *Cheyletus eruditus* (Schrank) are predators of stored-grain pests in China. The psocid *Liposcelis bostrychophila* Badonnel is prone to outbreaks in depots. To assess the potential of large-scale breeding with *Acarus siro* Linnaeus and the biological control potential of *C. malaccensis* and *C. eruditus* against *L. bostrychophila*, we determined the development times of different stages at 16, 20, 24, and 28 °C and 75% relative humidity (RH) while feeding on *A. siro,* as well as the functional responses of both species’ protonymphs and females to *L. bostrychophila* eggs at 28 °C and 75% RH. *Cheyletus malaccensis* had a shorter development time and longer adult survival time than *C. eruditus* at 28 °C and 75% RH and could establish populations faster than *C. eruditus* while preying on *A. siro*. The protonymphs of both species showed a type II functional response, while the females showed a type III functional response. *Cheyletus malaccensis* showed a higher predation ability than *C. eruditus*, and the females of both species had a higher predation ability than the protonymphs. Based on the observed development times, adult survival times, and predation efficiency, *Cheyletus malaccensis* has much greater biocontrol potential than *C. eruditus*.

## 1. Introduction

*Cheyletus malaccensis* Oudemans and *Cheyletus eruditus* (Schrank) (Acari: Cheyletidae) are predatory *Cheyletus* species that can adapt to a wide range of temperatures and humidity levels [1,2]. They are widely distributed in depots and feed commodity stores in Asia, Europe, and North America [3,4,5]. Both species prey on acarid grain mites and small arthropods, such as the eggs and larvae of stored-grain pests [6,7,8]. The two predatory mites present an environmentally friendly pest control method that is easy to use and does not rely on large instruments and equipment. Given these traits, the *Cheyletus* species can be employed to control stored products pests [9,10,11,12]. 

The booklouse *Liposcelis bostrychophila* Badonnel (Psocoptera: Liposcelididae) is a common stored-grain pest that is prone to outbreaks in depots [13,14]. It causes agricultural and economic losses, threatens food security [15], and is hard to control because of its small size, rapid reproduction, and ability to survive for long periods without food [14,16,17]. Current measures against *L. bostrychophila* still rely on insecticide application [18]. However, chemical pesticides have led to problems, such as pesticide residue, pest resistance, and risk of environmental pollution [19,20,21]. Therefore, it is important to develop alternative control methods for stored-grain pests [19]. Biological control, such as the use of predatory mites, is an effective alternative [22,23].

The functional response is one of the most important measures of predator behavior and can reveal different aspects of prey–predator interactions and assess predator predation efficacy by estimating the attack rate and handling time [24,25,26]. There are three types of functional responses: Holling I, II, and III [27]. The functional responses of *C. eruditus* and *C. malaccensis* to different prey have been reported separately, and most reports have shown the Holling II response. For *C. malaccensis*, the prey studied include *Aleuroglyphus ovatus* (Acari: Sarcoptiformes), *Lepidoglyphus destructor* (Acaridida: Glycyphagidae), *Megoura japonica* (Hemiptera: Aphididae), *Panonychus citri* (Acari: Tetranychidae), and so on [28,29,30,31,32]. For *C. eruditus*, the functional responses to *Tyrophagus putrescentiae* (Acari: Acaridae) and *A. ovatus* were studied [31,32]. The comparisons of the function responses of *C. malaccensis* and *C. eruditus* on the same prey have not been reported. There is also a lack of comparative studies on the life history and population parameters of these two cheyletids under the same feeding conditions.

In our previous work, two cheyletids preyed on the eggs and larvae of common stored-grain pests, especially *L. bostrychophila* [22,23]. In addition, *Acarus siro* Linnaeus (Acari: Acaridae) was proven to be a suitable feeding medium for the two species [22,23]. A recent sampling of stored-grain mites showed that the dominant predatory mites in grain depots have changed from *C. eruditus* to *C. malaccensis* in China [33,34,35,36]. There are various reasons for this shift in the dominant species, including population growth, prey range, and functional response. In such a situation, in order to choose a more suitable predatory mite to control stored grain pests in China in the future, the biological control potential of both mites needs to be evaluated and compared. Life history and functional response are two important evaluation criteria. In this study, we compared (1) the life history traits of the two mites on the feeding prey *A. siro* at 16, 20, 24, and 28 °C and 75% relative humidity (RH) to evaluate the potential of large-scale breeding with *A. siro* and (2) the functional responses of the two cheyletids on the eggs of *L. bostrychophila* to evaluate the predation efficiency. Based on the above information, we hope to provide fundamental information for effective large-scale artificial feeding and practical application.

## 2. Materials and Methods

### 2.1. Predator and Prey Sources

*Cheyletus malaccensis* was collected in Haikou, Hainan Province, China. *Cheyletus eruditus* was obtained from the Crop Research Institute, Prague, Czech Republic. Cheyletids were reared at 28 °C and 75% RH in the dark at the Institute of Grain Storage and Logistics Academy of National Food and Strategic Reserves Administration in Beijing, China. The Crop Research Institute provided the *A. siro*, and the mites were reared on whole wheat flour at 28 °C and 75% RH in the dark. *Liposcelis bostrychophila* was collected in Beijing (40°7′55.20″ N, 116°24′46.08″ E), China. Psocids were reared on a mixture of whole wheat flour and yeast (1:1) at 26 °C and 68% RH in the dark.

### 2.2. Life History Traits of Cheyletus malaccensis and Cheyletus eruditus

Female adult *C. malaccensis* and *C. eruditus* specimens were randomly selected from the feeding box. The mites were reared individually in plastic microrearing cells [37] with 15–20 *A. siro* adults as prey at 28 °C and 75% RH. After 24 h, 50 eggs were collected and designated as the F1 generation for further study. Eggs were individually placed inside microrearing cells and subjected to different temperatures (16, 20, 24, and 28 °C) at 75% RH in the dark. The data of *C. malaccensis* at 24 and 28 °C, with 75% RH come from our published articles [37]. Eggs were checked daily in each container. After the eggs hatched, 15 to 20 *A. siro* adults were added daily as food for each *C. malaccensis* and *C. eruditus*. Based on daily observations, the egg incubation periods and development times of larvae, protonymphs, deutonymphs (absent in males of *C. malaccensis*), and adult longevity (males and females) were determined. Twenty mites were included in each experiment for each predator mite species. *Cheyletus eruditus* can reproduce by gynogenesis, and males are rarely found [38,39]. Therefore, only the development times of *C. eruditus* females at different temperatures were determined. For *C. malaccensis*, the development times of both females and males were determined.

### 2.3. Functional Responses of Cheyletus malaccensis and Cheyletus eruditus 

All functional response experiments were conducted at 28 °C and 75% RH in the dark. Functional responses of protonymphs and female adults of *C. malaccensis* and *C. eruditus* to eggs of *L. bostrychophila* were investigated. Before an experiment, each predator was starved for 24 h to standardize the degree of hunger [40]. Protonymph and adult female predatory mites were provided with 1, 3, 5, 7, or 10 prey per rearing cell. After 24 h, predators were removed, and the number of eggs consumed was recorded. Each experiment included 15 individual predatory mites. 

The type of functional response curve was described by a logistic regression [41] as follows:(1)$$\frac{N_{a}}{N_{0}}=\frac{{\exp(P}_{0}{+P}_{1}N_{0}{+P}_{2}N_{0}^{2}{+P}_{3}N_{0}^{3})}{{1+\exp(P}_{0}{+P}_{1}N_{0}{+P}_{2}N_{0}^{2}{+P}_{3}N_{0}^{3})}$$
where *N_a_* is the number of prey consumed; *N*_0_ is the initial prey density; and *P*_0_, *P*_1_, *P*_2_, and *P*_3_ are the constant, linear, quadratic, and cubic coefficients, respectively. The type of functional response was determined by the values of *P*_1_ and *P*_2_, where significantly negative values (*P*_1_ < 0) exhibit a type II response and significantly positive values (*P*_1_ > 0) exhibit a type III response. Based on the logistical analysis results, the attack rat (*a*) and the handing time (*T_h_*) of type II were calculated using the Holling equation [42,43] as follows:*N_a_* = *a N*_0_/(1 + *a T_h_ N*_0_)(2)
where *N_a_* is the number of prey consumed; *a* (number of prey/predator) is the attack rate; *N*_0_ is the number of prey offered; and *T_h_* is the handling time in hours. Predation ability was determined as *a*/T*_h_*.

For type III, the following model [44] was used:*N_a_* = *N*_0_{1 − exp [*b N*_0_(*T_h_ N_a_* − 1)}(3)
where *b* is the attack rate for type III.

### 2.4. Statistical Analysis

Development times and adult longevity of different stages of *C. malaccensis* and *C. eruditus* were analyzed using IBM SPSS Statistics 20.0. One-way ANOVA and Tukey’s Honestly Significant Difference (HSD) multiple range tests were used with a significance level of 0.05. The means, standard errors, and variances of the population parameters were estimated using the bootstrap technique (10,000 samples) [45,46,47], which is contained in the TWOSEX-MSChart program [48,49,50]. Curve fitting was performed on functional response models using MATLAB (https://matlab.mathworks.com/, accessed on 1 April 2023).

## 3. Results

### 3.1. Life History of Cheyletus malaccensis and Cheyletus eruditus

Both *C. malaccensis* and *C. eruditus* completed their life cycles at temperatures ranging from 16 °C to 28 °C (Table 1, Figure 1). The incubation period of *C. eruditus* eggs ranged from 3.57 days (28 °C) to 11.5 days (16 °C). In *C. malaccensis*, for female eggs, the incubation period ranged from 2.25 days (28 °C) to 10.25 days (16 °C); for male eggs, it ranged from 1.83 days (28 °C) to 9.50 days (16 °C). The longest development times from egg to adult in both species were observed at 16 °C. For *C. eruditus* females, the development time was 49.38 days; for *C. malaccensis* females, it was 60.00 days; and for *C. malaccensis* males, it was 59.50 days. The shortest times were observed at 28 °C. For *C. eruditus* females, the development time was 15.71 days; for *C. malaccensis* females, it was 14.00 days; and for *C. malaccensis* males, it was 9.50 days. The development time from egg to adult for *C. malaccensis* was generally shorter than that of *C. eruditus* at the same temperature, except at 16 °C. The development times of the immature stages decreased with increasing temperature. In addition, the survival times of *C. malaccensis* were generally longer than those of *C. eruditus*, except at 16 °C. Female adult mites showed the most predatory potential. The survival times of *C. malaccensis* females were 58.00 days and 66.50 days at 24 and 28 °C, respectively; for *C. eruditus* females, the survival times were 19.50 days and 15.00 days at these temperatures. This represents about a three to four times difference in survival time. We also compared the population parameters of *C. malaccensis* and *C. eruditus* at different temperatures (Table 2). The net reproductive rate (*R*_0_) was significantly higher for *C. malaccensis* than *C. eruditus* at 24 and 28 °C. The fecundity of *C. malaccensis* was significantly higher than *C. eruditus* at 20, 24, and 28 °C. However, the intrinsic rate of increase (*r_m_*) and the finite rate of increase (λ) of *C. eruditus* were higher than those of *C. malaccensis* at 16 °C.

### 3.2. Functional Responses

The functional responses of *C. malaccensis* and *C. eruditus* to the eggs of *L. bostrychophila* at 28 °C and 75% RH are shown in Figure 2. The consumption of eggs of *L. bostrychophila* by *C. malaccensis* protonymphs and females was higher than that by *C. eruditus* (Table 3). In addition, the consumption of eggs by females was higher than that by protonymphs in both species. *C. malaccensis* protonymphs and *C. eruditus* protonymphs and females consumed more *L. bostrychophila* eggs than the first-instar larvae.

Based on logistic regression analyses (Table 4), the protonymphs of both species showed a type II functional response (*P*_1_ < 0). The females of *C. eruditus* exhibited a type II functional response (*P*_1_ < 0), while the females of *C. malaccensis* showed a type III functional response (*P*_1_ > 0, *P*_2_ < 0).

The parameters of the functional responses of *C. malaccensis* and *C. eruditus* are shown in Table 5. A comprehensive analysis of the attack rate (*a*) and handling time (*T_h_*) indicated that *C. malaccensis* had a higher predation ability than *C. eruditus*, and the females of both species had a higher predation ability than the protonymphs. The predation ability (*a*/T*_h_*) of *C. malaccensis* females was 9.43 on eggs, whereas that of *C. eruditus* females was 8.90 on eggs. For the protonymphs, the predation ability of *C. malaccensis* was 7.39 on eggs, whereas that of *C. eruditus* was 1.29 on eggs. We compared the predation ability between *C. malaccensis* and *C. eruditus* and found that the predation ability of *C. malaccensis* females was 1.06 times that of *C. eruditus* on eggs. For *C. malaccensis* protonymphs, the predation ability was 5.73 times that of *C. eruditus* on eggs. Comparing the predation ability of the same species, that of *C. malaccensis* females was 1.28 times that of the protonymphs on eggs. The predation ability of *C. eruditus* females was 6.90 times that of the protonymphs on eggs.

## 4. Discussion

The development times of *C. malaccensis* and *C. eruditus* fed *A. siro* at different temperatures were determined. Both species completed development from egg to adult at temperatures ranging from 16 °C to 28 °C. The development times of the two cheyletids decreased with increasing temperature. These trends are consistent with previous studies on the development of *C. malaccensis* and *C. eruditus* [1,51,52]. According to our previous studies, the most suitable temperature for reproduction in the two cheyletids is 28 °C [22,37]. In this study, the development times of *C. malaccensis* were shorter than those of *C. eruditus*, but the adult survival times of *C. malaccensis* were longer. The *R*_0_ and the fecundity of *C. malaccensis* were significantly higher than *C. eruditus* at 20, 24, and 28 °C, while there was no significant difference between these two species at 16 °C. It was reported that the *R*_0_ of *Neoseiulus californicus* (Acari: Phytoseiidae) were 23.67 at 20 °C, 28.56 at 25 °C, and 19.73 at 30 °C [53], which were lower than the two cheyletids. These differences may be caused by the different areas, and the storage environment is also more stable than the field. Therefore, with more rapid development, *C. malaccensis* could establish populations faster than *C. eruditus*, which increased its effectiveness as a biocontrol agent. However, the daily temperature fluctuations also significantly affected the development times and longevity of the biocontrol agents studied, resulting in marked deviations and potentially erroneous predictions when compared to their constant temperature regimen counterparts [54,55]. In practical application, we need to adjust measures according to the actual conditions, such as temperature and climate.

Functional response is one of the most important behavioral responses of predators, and it can be used to simulate prey–predator relationships and evaluate the potential predation efficiency of predators [26]. In this study, protonymphs and females of *C. malaccensis* showed different types of functional responses on *L. bostrychophila* eggs, that is, type II (protonymphs) and type III (females), while *C. eruditus* exhibited the same type II functional response. The reason for this may be that the same predator can reveal different functional responses depending on the size and stage of its prey, its degree of hunger, and so on [24,25,26].

In this study, both *C. malaccensis* and *C. eruditus* were potential predators on the eggs of *L. bostrychophila*. In both species, the most active stage was the adult female stage. *C. malaccensis* had higher attack rates than those of *C. eruditus*. The predation ability of *C. malaccensis* was 1.06 times (females preying on eggs) to 5.71 times (protonymphs preying on eggs) higher than that of *C. eruditus*. This result is consistent with that when *A. ovatus* larvae were used as prey, where a predation ability of 110.68 was observed for *C. malaccensis* females and a predation ability 9.27 was observed for *C. eruditus* females [28,32]. In addition, the daily maximum predation (1/T*_h_*) of *C. malaccensis* protonymphs was higher than that of *C. eruditus*, while this finding was the opposite in females. All these show that *C. malaccensis* was more effective in the control of *L. bostrychophila* and pest mites.

The functional response of a predator to its prey can be influenced by the prey stage and size, predator stage and size, speed of movement, and starvation level of predator [56,57,58]. In this study, *C. malaccensis* usually had a higher consumption than *C. eruditus*. However, the daily maximum predation on eggs of *C. malaccensis* females (5.67) was 4.11 times lower than that of *C. eruditus* (23.33). The reason for the lower consumption of eggs by *C. malaccensis* may be that it prefers dynamic and large prey, similar to *Amblyseius andersoni* Chant (Acari: Phytoseiidae) and *Euseius finlandicus* (Oudemans) (Acari: Phytoseiidae) [59,60]. Moreover, eggs may be less nutritious, and females need to increase their intake of energy and nutrition to reproduce. The large and fast prey was more difficult to hunt, as it required more time to attack and digest [61]. The eggs of *L. bostrychophila* are larger than those of *A. ovatus*. Therefore, the predation ability values on *A. ovatus* were higher than those of *L. bostrychophila*.

To improve the control of stored-grain pests and protect the environment, integrated pest management has attracted increasing attention [62]. This study was conducted by comparing the life histories of two predatory mites and the functional responses on *L. bostrychophila*. However, many factors affected the results of the functional responses observed in laboratory conditions and in practical applications, such as the temperature, the state of the predator and prey, and the degree of hunger. In future studies, we will also need to carry out a wider range of temperatures and attempt real-world assays of efficacy in the wheat depots of China. All of this will provide a theoretical basis and verification for the biological control of predatory mites.

## 5. Conclusions

In conclusion, *C. malaccensis* had a shorter development time and longer adult survival time than *C. eruditus* at 28 °C and 75% RH and could establish populations faster than *C. eruditus* while preying on *A. siro*. The protonymphs of both species showed a type II functional response, while females of *C. malaccensis* showed a type III functional response. Based on the observed development times, adult survival times, and predation efficiency, *C. malaccensis* has a much greater biocontrol potential than *C. eruditus* under laboratory conditions. Based on the above information, this study provided fundamental information for the effective large-scale artificial feeding and releasing program of the two cheyletids. All these studies will help to determine the effective temperatures and seasons for both rearing and releasing these predatory mites for different ecological areas of grain storage.

## Figures and Tables

**Figure 1 insects-14-00478-f001:**
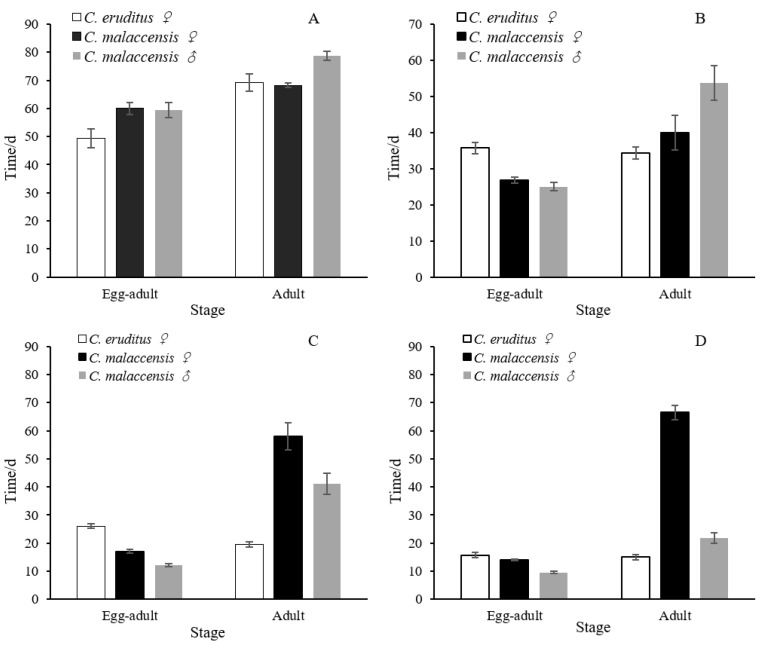
Egg to adult development times and adult survival times of *Cheyletus malaccensis* (males and females) and *Cheyletus eruditus* (females) at different temperatures: (**A**) 16 °C, (**B**) 20 °C, (**C**) 24 °C, and (**D**) 28 °C.

**Figure 2 insects-14-00478-f002:**
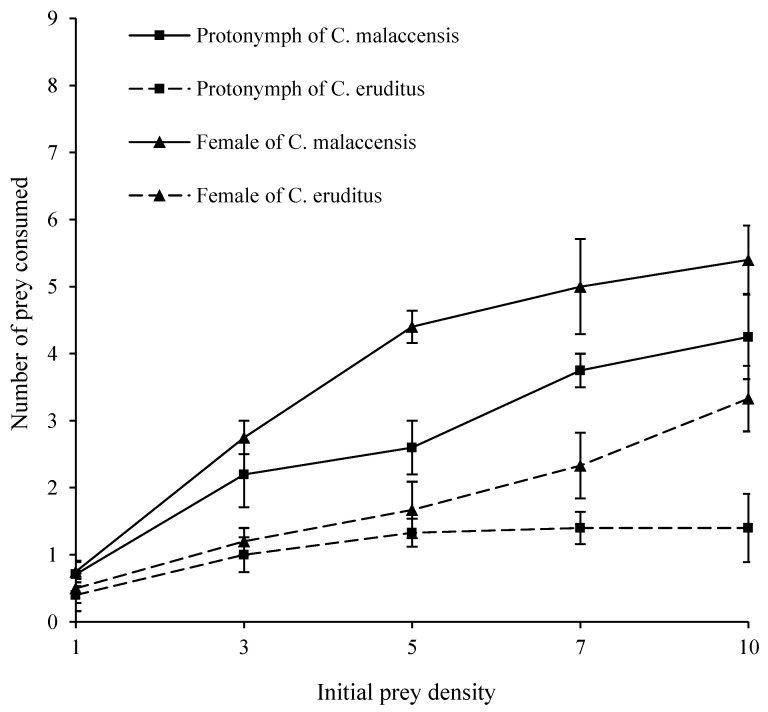
Functional responses of protonymphs and females of *Cheyletus malaccensis* and *Cheyletus eruditus* on eggs of *Liposcelis bostrychophila* at 28 °C and 75% RH.

**Table 1 insects-14-00478-t001:** Development times (days) of different stages of *Cheyletus eruditus* and *Cheyletus malaccensis* reared at different temperatures in the laboratory (mean ± SD).

Stage	Temperature (°C)	Predator		
		*C. eruditus*—Female	*C. malaccensis*—Female	*C. malaccensis*—Male
Egg	16	11.50 ± 1.34 Aa	10.25 ± 0.25 Aa	9.50 ± 0.50 Aa
	20	7.22 ± 0.52 Ab	4.83 ± 0.31 Ab	4.67 ± 0.33 Ab
	24	5.13 ± 0.64 Abc	2.57 ± 0.30 Bc	2.30 ± 0.15 Bc
	28	3.57 ± 0.43 Ac	2.25 ± 0.25 Bc	1.83 ± 0.40 Bc
Larva	16	14.00 ± 1.88 Aa	23.75 ± 1.03 Ba	27.25 ± 2.14 Ba
	20	7.78 ± 0.85 ABb	9.83 ± 0.40 BCb	13.00 ± 1.15 Cb
	24	7.50 ± 0.46 Ab	4.71 ± 0.29 Bc	4.80 ± 0.47 Bc
	28	4.14 ± 0.40 Ab	4.25 ± 0.16 Ac	4.17 ± 0.17 Ac
Protonymph	16	12.50 ± 1.45 Aa	9.25 ± 0.25 Aa	22.75 ± 1.49 Ba
	20	10.78 ± 0.85 Aa	5.17 ± 0.17 Bb	7.33 ± 0.33 Bb
	24	7.13 ± 0.64 Ab	4.86 ± 0.14 Bb	5.10 ± 0.38 Bbc
	28	4.14 ± 0.51 Ab	4.37 ± 0.26 Ac	3.50 ± 0.22 Ac
Deutonymph	16	11.38 ± 1.29Aa	16.75 ± 1.80 Aa	--
	20	10.00 ± 0.97 Aa	7.00 ± 0.26 Bb	--
	24	6.25 ± 0.45 Ab	4.86 ± 0.34 Bb	--
	28	3.86 ± 0.51 Ab	3.13 ± 0.23 Ab	--
Egg to adult	16	49.38 ± 3.32 Aa	60.00 ± 2.08 Aa	59.50 ± 2.75 Aa
	20	35.78 ± 1.56 Ab	26.83 ± 0.79 Bb	25.00 ± 1.15 Bb
	24	26.00 ± 0.82 Ac	17.00 ± 0.65 Bc	12.2 ± 0.53 Cc
	28	15.71 ± 0.97 Ad	14.00 ± 0.38 Ac	9.50 ± 0.43 Bc
Adult	16	69.25 ± 3.12 Aa	68.25 ± 0.75 Aa	78.75 ± 1.70 Aa
	20	34.33 ± 1.72 Ab	40.00 ± 4.73 Aba	53.67 ± 4.81 Bb
	24	19.50 ± 0.91 Ac	58.00 ± 4.77 Ba	41.10 ± 3.86 Cbc
	28	15.00 ± 0.98 Ac	66.50 ± 2.61 Ba	21.83 ± 1.85 Ac

Means followed by different lowercase letters indicate that there is significant difference at different temperatures at the same stage (*p* < 0.05), while means followed by different uppercase letters indicate that there is significant difference between *C. malaccensis* and *C. eruditus* at the same temperature, based on one-way ANOVA and Tukey’s HSD multiple range tests.

**Table 2 insects-14-00478-t002:** Population parameters of *Cheyletus malaccensis* and *Cheyletus eruditus* at different temperatures.

Population Parameters	Predatory	Temperature/°C			
		16	20	24	28
**T (day)**	** *C. malaccensis* **	70.67 ± 2.82 a *	38.62 ± 1.53 b	29.97 ± 1.97 c	24.43 ± 0.66 d *
	** *C. eruditus* **	48.85 ± 3.42 a *	37.01 ± 1.74 b	28.17 ± 0.67 c	17.39 ± 0.96 d *
** *R* ** ** _0_ ** **(offspring)**	** *C. malaccensis* **	9.91 ± 4.65 a	82.50 ± 28.59 b	204.61 ± 60.66 bc *	290.47 ± 70.38 c *
	** *C. eruditus* **	15.88 ± 3.54 a	46.60 ± 8.53 b	76.33 ± 11.39 c *	84.60 ± 18.49 bc *
** *r_m_* ** **(day^−1^)**	** *C. malaccensis* **	0.032 ± 0.008 a *	0.114 ± 0.012 b	0.178 ± 0.018 c	0.232 ± 0.015 d
	** *C. eruditus* **	0.057 ± 0.007 a *	0.104 ± 0.007 b	0.154 ± 0.007 c	0.255 ± 0.021 d
**λ (day^−1^)**	** *C. malaccensis* **	1.033 ± 0.009 a *	1.121 ± 0.013 b	1.194 ± 0.021 c	1.261 ± 0.018 d
	** *C. eruditus* **	1.058 ± 0.007 a *	1.109 ± 0.008 b	1.166 ± 0.008 c	1.291 ± 0.027 d
**Fecundity**	** *C. malaccensis* **	27.25 ± 8.01 a	165.00 ± 39.01 b *	526.14 ± 6.82 c *	544.63 ± 15.31 c *
	** *C. eruditus* **	18.14 ± 3.28 a	51.78 ± 7.80 b *	85.88 ± 7.85 c *	120.86 ± 8.89 d *

Means followed by different letters in the same line are significantly different at the 0.05 level according to the paired bootstrap test. * indicates that there is significant difference between the same population parameters of *C. malaccensis* and *C. eruditus* at the same temperature according to the paired bootstrap test at the 0.05 level. *R*_0_: net reproductive rate; T: mean generation time; *r_m_*: intrinsic rate of increase; λ: the finite rate of increase.

**Table 3 insects-14-00478-t003:** The consumption of *Cheyletus malaccensis* and *Cheyletus eruditus* preying on eggs of *Liposcelis bostrychophila* at 28 °C and 75% RH.

Stage of Predator	Predator	Prey Density				
		1	3	5	7	10
Protonymph	*C. malaccensis*	0.71 ± 0.18 a	2.20 ± 0.49 b *	2.60 ± 0.40 bc *	3.75 ± 0.25 cd *	4.25 ± 0.63 d *
	*C. eruditus*	0.40 ± 0.24 a	1.00 ± 0.26 ab *	1.33 ± 0.21 ab *	1.40 ± 0.24 b *	1.40 ± 0.51 b *
Female	*C. malaccensis*	0.75 ± 0.16 a	2.75 ± 0.25 b *	4.40 ± 0.24 c *	5.00 ± 0.71 c *	5.40 ± 0.51 c *
	*C. eruditus*	0.50 ± 0.22 a	1.20 ± 0.20 ab *	1.67 ± 0.42 ab *	2.33 ± 0.49 bc *	3.33 ± 0.49 b *

Means followed by different letters in the same line are significantly different at the 0.05 level according to the paired bootstrap test. * indicates that there is significant difference between the same stage of *C. malaccensis* and *C. eruditus* at the same prey density based on independent samples *t*-test at the 0.05 level.

**Table 4 insects-14-00478-t004:** The functional responses of *Cheyletus malaccensis* and *Cheyletus eruditus* based on logistic regression.

Stage of Predator	Predator	Parameter				*R* ^2^	Model
		*P* _0_	*P* _1_	*P* _2_	*P* _3_		
Protonymph	*C. malaccensis*	1.015	−0.007	−0.037	0.003	0.854	II
	*C. eruditus*	−0.295	−0.103	−0.012	0.001	0.998	II
Female	*C. malaccensis*	−0.487	1.961	−0.396	0.021	1.000	III
	*C. eruditus*	0.328	−0.361	0.04	−0.001	0.993	II

*P*_0_, *P*_1_, *P*_2_, and *P*_3_ are the maximum likelihood estimates (MLEs) of the intercept, linear, quadratic, and cubic coefficients, respectively. *P*_1_ < 0 indicates type II functional responses. *P*_1_ > 0 indicates type III.

**Table 5 insects-14-00478-t005:** Parameters of the functional responses of *Cheyletus malaccensis* and *Cheyletus eruditus* preying on eggs of *Liposcelis bostrychophila* at 28 °C and 75% RH.

Stage of Predator	Predator	Attack Rate (*a*)	Handling Time (T*_h_*)	Predation Ability (*a*/T*_h_*)	Daily Maximum Predation Amount (1/T*_h_*)	*R* ^2^
Protonymph	*C. malaccensis*	0.88	0.12	7.39	8.40	0.98
	*C. eruditus*	0.67	0.52	1.29	1.92	0.96
Female	*C. malaccensis*	1.66	0.18	9.43	5.67	0.95
	*C. eruditus*	0.38	0.04	8.90	23.33	0.99

*R*^2^ is the coefficient of determination. A high *R*^2^ suggests a good fit to the functional response model.

## Data Availability

Data may be requested from the corresponding authors.

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
