# Peer review of "Life Histories and Functional Responses of Two Predatory Mites Feeding on the Stored-Grain Pest Liposcelis bostrychophila Badonnel (Psocoptera: Liposcelididae)"

_insects, 2023, doi:10.3390/insects14050478_

Round 1

Reviewer 1 Report

The manuscript “Life histories and functional responses of two predatory mites feeding on the stored-grain pest Liposcelis bostrychophila Badonnel (Psocoptera: Liposcelididae)” by Weiwei Sun, Liyuan Xia and Yi Wu addresses an interesting topic, that is the implementation of biological control techniques for the control of this pest, an important problem of stored grain.

However, the manuscript is poorly elaborated and, at the present stage, it is not ready for publication. Following are my comments.

-The language used along the text is quite poor, without linking appropriately the ideas in a smooth rhythm that brings the reader to their conclusions. Also, the English needs a lot of improvement, with several sentences that have an awkward meaning, plurals not well done, etc. indicating that a professional correction is needed. The references cited do not always follow the consecutive ordinal order: citation of reference 10 is followed by number 15, and references 11, 12 13 and 14 are mentioned later.   

SIMPLE SUMMARY AND ABSTRACT

They are a copy-paste of one another. Authors should take advantage of these two sections for better expressing the ideas that they want to transmit to the reader and not repeat the same in the two sections.

KEY WORDS

“Liposcelis bostrychophila” and “functional response” are already mentioned in the title, no need to repeat them again in this section.

INTRODUCTION

The objective of the study has to be clearly defined together with the hypothesis that is tested. It is not clear why they want to know the biological parameters of the two predatory species, which is the purpose. Also, the question of the mass rearing with A. siro is not justified: is the present rearing media used for the rearing not working well? Which is the present media for the rearing? etc.

L-40, 48.- Include taxonomic affiliation of the species at first mention.

L-45.- Which environmentally friendly methods? Explain.

L-46-47.- Really? Include a reference that justify this statement.

L-55-56.- Again, if this is case, sustain the statement with references.

L-59.- Usually? This is not the appropriate expression.

L-65-67.- This is an awkward sentence. No idea of the meaning.

L-70-73.- Similarly, what is the point of this sentence? If you already know this, include a citation of the work.  

M & M

L-86.- If C. eruditus is present in China, why start a colony with individuals imported from the Czech Republic? Colonies of the other species used in the study (psocids, A. siro, A. malaccensis) were started with local individuals. Supposedly, colonies initiated with local individuals would be better adapted to Chinese environmental conditions.

L-95.-randomly selected? From where? 

L-96.- ‘15-20 A. siro as prey’. Which developmental stage was offered? Mixed stages?

L-98.- ‘Eggs were individually placed inside blocks’. What do you mean by ‘blocks’?

L-119.- is this figure necessary? I don’t see the point of including it.

RESULTS

L-151-154.- In these sentences, authors report data that indicate that females of C. eruditus developed faster than those of C. malaccensis both at 16ºC and at 28ºC.  

L-156-157.- But they conclude that C. malaccensis develops faster than C. eruditus. Here there is some mistake…. Also, to sustain such a statement, a statistical comparison between both species is needed. Looking at figure 2D, the development time from egg to adult of both species at 28ºC is very similar….

L-157-158.- Similarly, to sustain such a statement, a statistical comparison between the immature development parameters of both species is needed.

L-173-174.- Tables 2and 3 are a mess since numbers and letters do not fit well in the row space. Authors should think about a properly way of showing these data.

L-183-185.- Figure 2. Is it necessary to name each graph as A, B, etc. and simultaneously put the temperature at which experiments were done in the graph ? I think that only one type of notation is enough. Do the bars correspond to mean values? If this is so, the Standard Error of the mean should be included in each bar. The Y axis title ‘time/d’ what dies it mean? Days? Why the title of the X axis is also ‘time/d’? I just see developmental stages.

L-189-190.- Again, this statement needs to be supported by a statistical comparison.

L-195-196.- This sentence is incomplete.

L 219-220. Table 3. Daily maximum predation amount of female C. eruditus seems out of range. Is this an error?

(1/Th)

DISCUSSION

This section is merely a repetition of the results obtained, without a reasoning that interprets them point by point. Here authors have to address the objectives and the hypothesis posed in the Introduction.

Author Response

Dear Reviewer,

Thank you for your good suggestions. All the suggestions are very important and necessary for our manuscript (MS). Now we are turning to detailed respond to each comment.

-The language used along the text is quite poor, without linking appropriately the ideas in a smooth rhythm that brings the reader to their conclusions. Also, the English needs a lot of improvement, with several sentences that have an awkward meaning, plurals not well done, etc. indicating that a professional correction is needed. The references cited do not always follow the consecutive ordinal order: citation of reference 10 is followed by number 15, and references 11, 12 13 and 14 are mentioned later. 

Response: about the language, we have invited the English polishing agency recommended by the journal to improve it. We also rearranged the order of cited references.

SIMPLE SUMMARY AND ABSTRACT

They are a copy-paste of one another. Authors should take advantage of these two sections for better expressing the ideas that they want to transmit to the reader and not repeat the same in the two sections.

Response: We have rewritten the simple summary according to your suggestion.

KEY WORDS

“Liposcelis bostrychophila” and “functional response” are already mentioned in the title, no need to repeat them again in this section.

Response 1: We have deleted “Liposcelis bostrychophila” and “functional response” in the keywords.

INTRODUCTION

The objective of the study has to be clearly defined together with the hypothesis that is tested. It is not clear why they want to know the biological parameters of the two predatory species, which is the purpose. Also, the question of the mass rearing with A. siro is not justified: is the present rearing media used for the rearing not working well? Which is the present media for the rearing? etc.

Response: we revised the objective of the study. ‘the dominant predatory mites in grain depots have changed from C. eruditus to C. malaccensis in China. In such a situation, in order to choose more suitable predatory mite to control stored grain pests in China biological control potential of the both mites need to be evaluated and compared. So we compared the life history and functional response.

the question of the mass rearing with A. siro, we use wheat bran to feed A. siro, and A. siro to feed predatory mites, which was proven to be the suitable feeding medium of the two species.

Point: L-40, 48.- Include taxonomic affiliation of the species at first mention.

Response: we have revised it according to your suggestion. We have added all of the taxonomic affiliation of the species at first mention.

Point: L-45.- Which environmentally friendly methods? Explain.

Response: Yes, the sentence was not expressed accurately enough, we had rewritten it ‘The two predatory mites was an environmentally friendly pest control method that was easy to use and did not rely on large instruments and equipment.’

Point: L-46-47.- Really? Include a reference that justify this statement.

Response: changed the sentenceGiven these traits, the Cheyletus species are widely employed to control stored products pests.” into “Given these traits, the Cheyletus species can be employed to control stored products pests.” and added the relevant references.

Point: L-55-56.- Again, if this is case, sustain the statement with references.

Response: we added the references.

Point: L-59.- Usually? This is not the appropriate expression.

Response: we deleted ‘usually’.

Point: L-65-67.- This is an awkward sentence. No idea of the meaning.

Response: we rewrote these sentences as following.

 ‘The comparison of function responses of C. malaccensis and C. eruditus on the same prey have not been reported. Also there is a lack of comparative studies on the life history and population parameters of the two cheyletids under the same feeding conditions.’

Point: L-70-73.- Similarly, what is the point of this sentence? If you already know this, include a citation of the work.

Response: we rewrote these sentences and add the references as following.

‘In our previous work, two cheyletids preyed on the eggs and larvae of common stored-grain pests, especially L. bostrychophila [33]. In addition, Acarus siro Linnaeus (Acari: Acaridae) was proven to be the suitable feeding medium of the two species[34].’

M & M

Point: L-86.- If C. eruditus is present in China, why start a colony with individuals imported from the Czech Republic? Colonies of the other species used in the study (psocids, A. siro, A. malaccensis) were started with local individuals. Supposedly, colonies initiated with local individuals would be better adapted to Chinese environmental conditions.

Response: We started the study of predatory mites in 2013 by an international cooperation project with the Czech Republic. The initial data were all based on the Czech strain. When we collected Chinese strains, we compared between Czech strain and Chinese strains, and found that there was no difference.

Point: L-95.-randomly selected? From where?

Response: changed it into ‘Female adult C. malaccensis and C. eruditus were randomly selected from feeding box.’

Point: L-96.- ‘15-20 A. siro as prey’. Which developmental stage was offered? Mixed stages?

Response: changed ‘with 15-20 A. siro as prey’ into ‘with15-20 A. siro adults as prey.’

Point: L-98.- ‘Eggs were individually placed inside blocks’. What do you mean by ‘blocks’?

Response: blocks were the plastic micro rearing cells. Changed ‘blocks’ into ‘micro rearing cells’

Point: L-119.- is this figure necessary? I don’t see the point of including it.

Response: deleted this figure.

RESULTS

Point: L-151-154.- In these sentences, authors report data that indicate that females of C. eruditus developed faster than those of C. malaccensis both at 16ºC and at 28ºC.

L-156-157.- But they conclude that C. malaccensis develops faster than C. eruditus. Here there is some mistake…. Also, to sustain such a statement, a statistical comparison between both species is needed. Looking at figure 2D, the development time from egg to adult of both species at 28ºC is very similar….

Point: L-157-158.- Similarly, to sustain such a statement, a statistical comparison between the immature development parameters of both species is needed.

Point: L-173-174.- Tables 2and 3 are a mess since numbers and letters do not fit well in the row space. Authors should think about a properly way of showing these data.

Response: Yes, these sentences were not expressed accurately. We rewrote the sections and rearranged the tables. In order to display and comparison clearly, we had integrated three tables into one table. Also we added a statistical comparison between the immature development parameters of both species in Table 2.

Point: L-183-185.- Figure 2. Is it necessary to name each graph as A, B, etc. and simultaneously put the temperature at which experiments were done in the graph ? I think that only one type of notation is enough. Do the bars correspond to mean values? If this is so, the Standard Error of the mean should be included in each bar. The Y axis title ‘time/d’ what dies it mean? Days? Why the title of the X axis is also ‘time/d’? I just see developmental stages.

Response: we have revised the Figure according to your suggestion.

Point: L-189-190.- Again, this statement needs to be supported by a statistical comparison.

Response: we have revised it according to your suggestion. We have added the statistical comparisons of the consumption in Table 3.

Point: L-195-196.- This sentence is incomplete.

Response: we had revised it.

Point: L 219-220. Table 3. Daily maximum predation amount of female C. eruditus seems out of range. Is this an error?

Response 19: we checked data. It was that. Daily maximum predation (1/ Th ) amount of female C. eruditus was too high. It was mainly because Handling time (Th) of C. eruditus very short. This is just a theoretical value, in practical application, factors such as the total feeding time and the consumption will also be considered.

DISCUSSION

Point: This section is merely a repetition of the results obtained, without a reasoning that interprets them point by point. Here authors have to address the objectives and the hypothesis posed in the Introduction.

Response: we revised the discussion according to your suggestion.

Reviewer 2 Report

The authors have graphed and presented their results clearly, drawing some attention to the implications of their findings. I found the study of interest and a good contribution to the knowledge of bio ecology of insect biocontrol agents (BCAs). The methods used are appropriate for the objectives of the work and, in general, well depicted. The resulting tales are sufficient and informative, helping to follow the reasoning throughout the manuscript. The Insects journal is perhaps appropriate, but I suggest resubmitting the work once the following corrections are made.

Firstly, the Intro and Discussion provide no insight on how this MS relates to the various other ones cited in the text or concerns that have been raised by other researchers. The authors do not present any hypotheses or expectations that could be connected to previous studies; adding these details will improve the paper as indicated below in my comments. The authors should also clearly explain why the study was done, why it was important, and how it fits with other studies. It should be clear and concise. The intro should also include what outcome(s) they expect, and how it would help support or refute their hypotheses or answer their questions.

However, my primary concern is that the authors are extrapolating the applicability of their results beyond what the design supports. These are only data from a set of four highly artificial constant laboratory conditions (i.e., 16C-28C), so the inference power of the paper is very limited, but authors do not acknowledge this detail at all and need to be more forthcoming. The effect of fluctuating temperature profiles on BCAs was not investigated in this study. This is a critical limitation of the study, and the authors must concede and discuss this. The interaction of cyclic temperatures with nonlinear characteristics of BCA development curves can introduce significant deviations from the results obtained here, and especially at the lower and higher temperatures of development functions which were not investigated in this study (e.g., <16C and >28C). Studies across a broader set of fluctuating temperature regimes are therefore encouraged so that more realistic effect of temperature on biological parameters of BCAs could be elucidated, as this is the closest to temperature fluctuations that occur in the field. So, I am suggesting to the authors to tone-down the language a little and admit that there are still substantive uncertainties to be considered.

Some of the authors’ statements would be much stronger if they tie their work to the body of literature that has built up on the bio ecology of BCAs in California, e.g., Journal of Economic Entomology 112:1560-1574 and Journal of Economic Entomology 112:1062-1072. These studies provide strong evidence that daily temperature fluctuations significantly affected development times and longevity of BCAs studied, resulting in marked deviations and potentially erroneous predictions when compared to their constant temperature regimen counterparts. In these studies, each fluctuating temperature profile was modeled after field recorded temperatures that had the desired average target temperature. These are the first studies ever to undergo such analysis. This article should provide details on all these fronts to provide the proper context for the work. This is not to diminish the data gathered in this study, as they are of value. But it is important for the authors not to overgeneralize, and to warn the reader, including regulatory agencies, against doing so as well. Adding these details will improve the discussion.

Also, the discussion lacks real concluding remarks in my opinion, and if I was a BC practitioner or consultant, I’d want to see these recommendations for my area or city.

Good luck!

Author Response

Dear Reviewer,

Thank you for your good suggestions. All the suggestions are very important and necessary for our manuscript (MS). Now we are turning to detailed respond to each comment.

Point 1: Intro and Discussion provide no insight on how this MS relates to the various other ones cited in the text or concerns that have been raised by other researchers. The intro should also include what outcome(s) they expect, and how it would help support or refute their hypotheses or answer their questions.

Response: we have revised it according to your suggestion.

Line 91-94: We added the sentence “In such a situation, in order to choose a more suitable predatory mite to control stored grain pests in China in the future, the biological control potential of the both mites needs to be evaluated and compared.”

Line 99-101: We added the sentence “Based on the above information, we hope to provide fundamental information for effective large-scale artificial feeding and practical application.”

Line 287-288: We added the sentence “In practical application, we need to adjust measures according to the actual conditions such as temperature and climate.”

Line 327-332: We added the sentence “However, many factors affected the results of the functional responses observed in laboratory conditions and in practical application, such as temperature, the state of the predator and prey, the degree of hunger, and so on. In future studies, we will also need to carry out a wider range of temperatures and attempt real-world assays of efficacy in the wheat depots of China. All of this will provide a theoretical basis and verification for the biological control of predatory mites.”

Point 2: Studies across a broader set of fluctuating temperature regimes are therefore encouraged so that more realistic effect of temperature on biological parameters of BCAs could be elucidated, as this is the closest to temperature fluctuations that occur in the field. So, I am suggesting to the authors to tone-down the language a little and admit that there are still substantive uncertainties to be considered.

Response 2: we have revised it according to your suggestion. In the unpublished data, we calculated the theoretical maximum temperature and minimum temperature based on six temperature. In this manuscript, we have revised the language.

Point 3: Some of the authors’ statements would be much stronger if they tie their work to the body of literature that has built up on the bio ecology of BCAs in California. This article should provide details on all these fronts to provide the proper context for the work. This is not to diminish the data gathered in this study, as they are of value. But it is important for the authors not to overgeneralize, and to warn the reader, including regulatory agencies, against doing so as well. Adding these details will improve the discussion.

Response 3: we have revised it according to your suggestion. We have added the references involved, such as data of R0 of Neoseiulus californicus.

Point 4: The discussion lacks real concluding remarks

Response 4: we have revised it according to your suggestion.

327-332: We added the sentence “However, many factors affected the results of the functional responses observed in laboratory conditions and in practical application, such as temperature, the state of the predator and prey, the degree of hunger, and so on. In future studies, we will also need to carry out a wider range of temperatures and attempt real-world assays of efficacy in the wheat depots of China. All of this will provide a theoretical basis and verification for the biological control of predatory mites.”

Line 340-342: We changed the sentence “Based on the observed development times, adult survival times, and predation efficiency, C. malaccensis has much greater biocontrol potential than C. eruditus.” into “Based on the observed development times, adult survival times, and predation efficiency, C. malaccensis has much greater Line

Line 342-346: We added the sentence “Based on the above information, this study provided fundamental information for the effective large-scale artificial feeding and releasing program of the two cheyletids. All of these studies will help to determine the effective temperatures and seasons for both rearing and releasing these predatory mites for different ecological areas of grain storage.”

Reviewer 3 Report

Dear Authors,

The manuscript is interesting. Please find my comments in the attached file.

Best regards

Dear Authors,

English language was good, but there are things that need to be changed. Please find my comments in the attached file.

Best regards

Author Response

Dear Reviewer,

Thank you for your good suggestions. All the suggestions are very important and necessary for our manuscript (MS). Now we are turning to detailed respond to each comment.

Point 1: Lin14 different stages means which stages?

Response 1: It means the immature stages and adults.

Point 2: did you experiment with Acarus siro apart from Liposcelis bostrychophila? If yes please change the title

Response 2: We didn’t carry out development time of the two cheyletids with Liposcelis bostrychophila.

Point 3: L36-37.- delete.

Response 3: we have revised it according to your suggestion.

Point 4: L-40-41.- please add the order and the family of these species

Response 4: we have revised it according to your suggestion. We have added all of the taxonomic affiliation of the species throughou the manuscript.

Point 5: Article format

Response 5: we have revised it according to your suggestion.

Point 6: L-98- how many eggs did you used for each temperature?

Response 6: we have revised it according to your suggestion. 50 eggs were collected and designated as the F1 generation for further study.

Point 7: please write "days" not "d". please change throughout the manuscript

Response 7: we have revised it according to your suggestion. We have changed “d” into “days” throughout the manuscript.

Point 8: Table 2,3 is hard to read, please make it clear

Response 8: we have revised it according to your suggestion. We have rearranged the tables. In order to display and comparison clearly, we had integrated Table 1 and Table 2 into one table.

Point 9: please unify the discussion into one text. also enhance it.

Response 9: we have revised it according to your suggestion. We have unified the discussion into one text.

Point 10: L 232-234- please enhance the discussion with life history from other stored product insect or mite species. You can add some information about the fecundity, developmental time etc in comparison with temperature. It is rather small section, make sure you discuss every result you find

Response 10: we have revised it according to your suggestion. We have added the references involved, such as data of R0 of Neoseiulus californicus.

Point 11: L-241-242.- please add references

Response 11: we have revised it according to your suggestion. We have added the references involved.

Point 12: L-259-260.- please add references and enhance this part too, it is too short

Response 12: we have revised it according to your suggestion. We have added the references involved.

Point 13: L-264-271.- please add more references and enhance this part.  some of these sentences should be in the conclusion section, not in the discussion.

Response 13: We have rewritten this part.

327-332: We added the sentence “However, many factors affected the results of the functional responses observed in laboratory conditions and in practical application, such as temperature, the state of the predator and prey, the degree of hunger, and so on. In future studies, we will also need to carry out a wider range of temperatures and attempt real-world assays of efficacy in the wheat depots of China. All of this will provide a theoretical basis and verification for the biological control of predatory mites.”

Point 14: L-273-275- This sentence is missing something, is like you coried it from the introduction and abstract

Response 14: We have rewritten this part and enhanced it according to your suggestion.

Line 342-346: We added the sentence “Based on the above information, this study provided fundamental information for the effective large-scale artificial feeding and releasing program of the two cheyletids. All of these studies will help to determine the effective temperatures and seasons for both rearing and releasing these predatory mites for different ecological areas of grain storage.”

Point 15: L-277-281- what can the world get from this study? How can this be useful?

Response 15: we have revised it according to your suggestion.

Line 340-342: We changed the sentence “Based on the observed development times, adult survival times, and predation efficiency, C. malaccensis has much greater biocontrol potential than C. eruditus.” into “Based on the observed development times, adult survival times, and predation efficiency, C. malaccensis has much greater Line

Line 342-346: We added the sentence “Based on the above information, this study provided fundamental information for the effective large-scale artificial feeding and releasing program of the two cheyletids. All of these studies will help to determine the effective temperatures and seasons for both rearing and releasing these predatory mites for different ecological areas of grain storage.”

Round 2

Reviewer 2 Report

Authors have done a nice job addressing all of my original comments. Thank you.